# Effects of the Large-Scale Circulation on Temperature and Water Vapor Distributions in the Π Chamber

Jesse C. Anderson[1], Subin Thomas[1], Prasanth Prabhakaran[1], Raymond A. Shaw[1], and Will Cantrell[1]

[1]Michigan Technological University, 1400 Townsend Drive, Houghton, MI/USA.

**Correspondence:** Will Cantrell (cantrell@mtu.edu)

**Abstract.**

Microphysical processes are important for the development of clouds and thus Earth's climate. For example, turbulent fluctuations in the water vapor mixing ratio, $r$, and temperature, $T$, cause fluctuations in the saturation ratio, $S$. Because $S$ is the driving factor in the condensational growth of droplets, fluctuations may broaden the cloud droplet size distribution due to individual droplets experiencing different growth rates. The small scale turbulent fluctuations in the atmosphere that are relevant to cloud droplets are difficult to quantify through field measurements. We investigate these processes in the laboratory, using Michigan Tech's Π Chamber. The Π Chamber utilizes Rayleigh-Bénard convection (RBC) to create the turbulent conditions inherent in clouds. In RBC it is common for a large scale circulation (LSC) to form. As a consequence of the LSC, the temperature field of the chamber is not spatially uniform. In this paper, we characterize the LSC in the Π chamber and show how it affects the shape of the distributions of $r$, $T$ and $S$. The LSC was found to follow a single roll with an updraft and downdraft along opposing walls of the chamber. Near the updraft (downdraft), the distributions of $T$ and $r$ were positively (negatively) skewed. At each measuring position $S$ consistently had a negatively skewed distribution , with the downdraft being the most negative.

*Copyright statement.* TEXT

## 1 Introduction

The effects that clouds have on Earth's climate system are quite sensitive to the details of processes that occur on scales much smaller than the cloud as a whole. For example, two clouds with the same amount of liquid water can behave differently depending upon their droplet size distributions. If the liquid water content (LWC) is distributed over a large number of small droplets the cloud will be quite reflective and unlikely to precipitate. Conversely, a cloud with the the same amount of liquid water distributed over fewer droplets will be less reflective and more likely to precipitate (Twomey, 1977; Albrecht, 1989; Pincus and Baker, 1994).

The two principal processes that shape the cloud droplet size distribution are condensation/evaporation and collision-coalescence. Condensation is driven by gradients in the saturation ratio, $S \equiv \frac{e}{e_s}$, between the environment and the surface

of the droplets and can result in a rapid increase in size for small droplets. Here, $e$ is the water vapor partial pressure and $e_s$ is the saturation vapor pressure. However, because $\frac{dR}{dt} \propto \frac{S-1}{R}$, where $R$ is the radius of the droplet and $t$ is time, growth to sizes larger than $R \approx 10\ \mu$m takes longer than the typical lifetime of most clouds (Grabowski and Wang, 2013). On the other hand, the rate of collision-coalescence only becomes appreciable once some droplets reach a size of $R \approx 20\ \mu$m (Pruppacher and Klett, 1997, Chpt.13). How this gap in size between growth by condensation and collision-coalescence is bridged has been one of the enduring questions in cloud physics for the past few decades (Grabowski and Wang, 2013).

Clouds are ubiquitously turbulent, which has been suggested as a mechanism for broadening the cloud droplet size distribution. Turbulence may increase the likelihood of collisions between droplets (Wang et al., 2008). Fluctuations in water vapor concentration and temperature due to turbulence also result in fluctuations in the saturation ratio. The variance in $S$ implies each droplet in a cloud experiences a different growth rate, and the differing growth rates could broaden the cloud droplet size distribution (Gerber, 1991; Korolev and Isaac, 2000; Chandrakar et al., 2016; Desai et al., 2018). However, in the atmosphere it is difficult to quantify the fluctuations in $S$. (See Siebert and Shaw (2017) for one example.) A laboratory setting, where the effects of fluctuations in $S$ on activation and the drop size distribution can be quantified (Chandrakar et al., 2016, 2017, 2020b; Prabhakaran et al., 2020), is one way some of the enduring questions associated with growth of cloud droplets can be addressed. (Laboratory investigations of the effect of turbulence on collision-coalescence would likely require facilities with greater vertical extents than are currently available (Shaw et al., 2020).) The laboratory environment has the benefit that clouds can be formed and sustained repeatably under known boundary conditions and their properties measured in steady state conditions, which allows ample time for statistically meaningful measurements.

The laboratory facility is Michigan Tech's $\Pi$ Chamber, described in Chang et al. (2016). We provide a brief overview here. The chamber operates under conditions of turbulent Rayleigh-Bénard convection (RBC), where the lower surface of the cell is set to a higher temperature than the upper surface. These conditions cause turbulent mixing due to the buoyancy difference between warm and cool air. As the air mixes it results in fluctuations in the temperature; the nature of these fluctuations in the scalar field have been studied intensely (see *e.g.* Ahlers et al., 2009; Chillà and Schumacher, 2012). As examples, fluctuations in temperature have been shown to depend on the geometry of the convection cell, the intensity of turbulence and the working fluid.

The turbulent intensity and fluid properties are typically described using the Rayleigh number ($Ra = \frac{\alpha g H^3 \Delta T}{\kappa \nu}$) and the Prandtl number ($Pr = \frac{\nu}{\kappa}$) respectively, where $g$ is the acceleration due to gravity, $\Delta T$ is the temperature difference between the top and bottom plates separated by distance $H$, $\alpha$ is the coefficient of thermal expansion of the fluid, $\kappa$ is its thermal diffusivity, and $\nu$ is its kinematic viscosity. For a gradient in both temperature and water vapor, the Rayleigh number becomes (Niedermeier et al., 2018)

$$Ra = \frac{\alpha g H^3 \Delta T}{\kappa \nu} + \frac{g \epsilon \Delta r H^3}{\kappa \nu} \tag{1}$$

where $\epsilon \equiv \frac{m_d}{m_v} - 1$, $m_d$ is the molecular mass of dry air, $m_v$ is the molecular mass of water, and $r$ is the vapor mixing ratio. Note that for the range of conditions explored in this paper, $Ra$ is dominated by the first term in Eq. (1). Studies of the temperature profile (statistics) on the vertical axis of cylindrical cells show a well mixed fluid with little gradient outside the boundary

layer (Belmonte and Libchaber, 1996; Sakievich et al., 2016; Xie et al., 2019). There are fewer studies of the off-center bulk temperature profiles (statistics) (Liu and Ecke, 2011; He et al., 2019).

In turbulent Rayleigh-Bénard convection a structure forms in the fluid flow, referred to as the "mean wind of turbulence" or large-scale circulation (LSC). It is a mean, background flow within the overall turbulent motion in the chamber. For cells that have an aspect ratio ($\Gamma \equiv D/H$, where $D$ is the cell diameter) near one or two, the LSC usually takes the form of a single roll which spans the diameter of the chamber (Xia et al., 2008). This single roll has an updraft on one side of the cell that has a positive mean vertical velocity and a higher temperature than the center of the chamber. Along the opposite side of the cell
the fluid typically has a negative vertical velocity and lower temperatures. A visualization of the circulation is shown on the left side of Fig 1. For cells with $\Gamma \gtrsim 4$, multiple convective rolls become the dominant circulation mode (Xia et al., 2008). We anticipate the circulation in the $\Pi$ chamber will follow a single roll due to the chamber having $\Gamma = 2$.

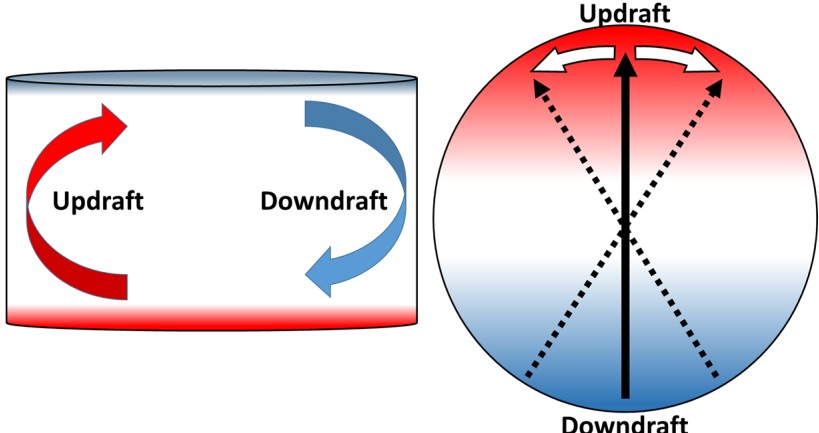

**Figure 1.** Cross section (left) and plan view (right) of the motion of the LSC in a cylindrical geometry of aspect ratio 2. On the left figure the arrows indicate the mean direction of the airflow due to the circulation with the red arrow describing the warm updraft and the blue arrow representing the cool downdraft. On the right side of the figure the black solid arrow points towards the updraft. The white arrows and dotted black arrows describe the azimuthal oscillations in the circulation.

   The updraft-downdraft associated with the large scale circulation typically adopts a specific orientation within the cell, but also has several oscillatory modes about that mean position. One of the primary oscillations is azimuthal, about a vertical axis
that runs through the center of the cell (Brown and Ahlers, 2007b). Often the azimuthal oscillation at the top and bottom of the chamber are out of phase. The resulting oscillation is called the torsional mode (Funfschilling et al., 2008). In addition the LSC has been shown to oscillate side to side in what has been referred to as the sloshing mode (Brown and Ahlers, 2009). In cells with very high symmetry, the LSC can spontaneously cease and reorient to a different angular location (Brown et al., 2005; Brown and Ahlers, 2006, 2007a; Xi et al., 2006). An asymmetry, such as tilting the cell, can fix the orientation of the LSC (Xi
et al., 2009).

One of the primary motivations for studies in the Π chamber is to understand cloud microphysical processes in the atmosphere; one emphasis is determining how fluctuations in the saturation ratio affect the drop size distribution and aerosol activation. For example in Chandrakar et al. (2016), a zeroth order model (a stochastic differential equation) was used to quantify the effect of fluctuations in $S$ on droplet growth. The treatment of fluctuations in temperature and water vapor concentration in the chamber was recently refined using a one dimensional turbulence model (ODT) (Chandrakar et al., 2020a), which incorporates vertical variations. It should also be noted that a mean field approach captures many aspects of the microphysical processes in the chamber (Krueger, 2020). While these models have provided valuable insights into processes in the chamber, the assumption of no spatial variability or of variability in only the vertical direction comes into question in the presence of an LSC in the chamber, where the mean temperature is horizontally nonuniform. It is necessary to measure the spatial and temporal variability in $r$, $T$ and $S$ in order to determine how closely the models of reduced dimensionality capture the true variability in the chamber.

In this paper we describe the basic characteristics of the flow in the chamber, including the large scale circulation because of the importance of these quantities on the distribution of temperature and water vapor, and thus on the saturation ratio. While measurements of temperature in turbulent Rayleigh-Bénard convection are ubiquitous, as noted above, measurements of water vapor concentration are rare, and differ in some fundamental aspects from measurements of temperature. We first describe how we compare measurements of water vapor and temperature through an exploration of how a path averaged measurement differs from an ideal point measurement. Next, we describe the behavior of the LSC in the chamber across several temperature differences. We then describe how the LSC changes the shape of the temperature distributions in the bulk of the chamber. Finally we present measurements of water vapor concentration, temperature and the saturation ratio, $S$, at different locations in the LSC of the Π-chamber for both dry ($S < 1$) and moist ($S > 1$) convection.

## 2 Methods

### 2.1 Facility

Our experiments were conducted in Michigan Tech's Π Chamber with the cylindrical insert in place; in those conditions, $\Gamma = 2$. The cylindrical insert restricts the volume of the chamber to 3.14 m$^3$. To induce convection, the top and bottom control surfaces within the chamber are set such that $T_{Top} < T_{Bottom}$ and $T_{Sidewall} = (T_{Top} + T_{Bottom})/2$. In the experiments reported here, data was recorded for temperature differences ($\Delta T = T_{Bottom} - T_{Top}$) up to 16 K. These measurements are recorded at 1 Hz. The chamber is described in greater detail in Chang et al. (2016).

We present measurements in two different conditions in the chamber; dry and moist convection. In our experiments the distinction between dry and moist convection is determined by the saturation ratio, $S$, defined as

$$S \equiv \frac{r}{r_s(T)} = \frac{e}{e_s(T)} \tag{2}$$

where $r$ is the water vapor mixing ratio, $r_s(T)$ is the saturated mixing ratio which is a function of the temperature $T$, $e$ is the vapor pressure and $e_s$ is the saturation vapor pressure. In practice, the saturation values are calculated from an empirical

approximation of the Clausius-Clapeyron equation, in our case the Magnus approximation, using the measured value of $T$ (Lamb and Verlinde, 2011). We define dry convection as a subsaturated condition ($S < 1$) in the chamber. In moist convection the chamber is supersaturated ($S > 1$), with the bottom, top, and sidewalls of the chamber being saturated. In moist conditions, a cloud would form if aerosol particles were present, but for these experiments, we did not inject aerosols into the chamber which prevents the formation of cloud droplets.

## 2.2 Instrumentation

Our basic temperature measurement is through 100 $\Omega$, thin film, platinum resistance thermometers (RTDs, Minco, S17624, 100 $\Omega \pm 0.12\%$). After calibrating the RTDs against each other in an isolated box, the difference between two RTDs was then calculated. The uncertainty was then calculated by taking the standard deviation of that difference, which was determined to be $\pm 0.001$K. A ring of eight RTDs, 1 cm away from the wall of the cylinder was used to determine the orientation and amplitude of the large scale circulation. The RTDs are on the horizontal midplane of the chamber and are evenly spaced such that the angular distance between each one is $\pi/4$ radians. The setup for this experiment can be seen in Fig. 2.

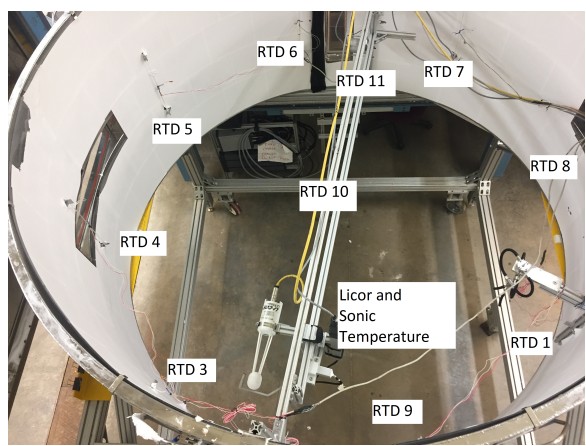

**Figure 2.** Photo showing the sensor locations (from above) for the moist convection experiments. The RTDs are 1 cm from the wall. The LiCor and sonic temperature sensor are collocated on the traverse that spans the chamber (see bottom of figure). RTD 10 is in the center of the chamber and is used for calibration of the sonic temperature. The diameter of the cylinder is 2 m.

As noted above, the primary measurement in Rayleigh-Bénard convection has been temperature, which can be measured with a variety of sensors with the required accuracy and precision. Quantitative measurement of concentration is much less common. Measurements of water vapor concentration are limited by the dynamic range of the sensor and temporal resolution (*e.g.* capacitance hygrometers) or by the path over which the measurement is averaged (*e.g.* absorption hygrometers). We use a LiCor LI-7500A infrared hygrometer at 1 Hz to measure water vapor, with a 5 Hz averaging time. The path length, $d$, of the LiCor is 12.5 cm with a measurement volume of $\approx 12$ cm$^3$. To ensure that we measure $T$ with the same spatial and similar temporal resolution as $r$ (to get a reliable value of $S$) we use a high speed sonic temperature sensor (Applied Technologies,

Inc.); it has a path length of ≈ 13 cm and was set to sample at 1 Hz, with a 1 Hz averaging time. The sonic temperature sensor operates on the same physical principle as a sonic anemometer, using the transit time of an acoustic signal in order to calculate the speed of sound, which is a function of the temperature and humidity. In this case, the temperature sensor is sensitive to the virtual temperature, $T_v \simeq (1 + 0.61r)T$ (Lamb and Verlinde, 2011), which can be converted to the actual temperature using measured water vapor concentrations from the LiCor.

Both water vapor and temperature sensors were collocated on a traverse system so that they measured roughly the same volume. The traverse allowed us to move the sensors along a line that bisects the chamber. The three measurement points on traverse lie on a 5 cm offset from the line that bisects the chamber, with one near the center and two on opposing sides of the chamber. The two off-center locations will be referenced as the updraft and downdraft later in the paper and are about 20 cm from the nearest sidewall. The traverse system is located on the horizontal mid-plane ($z = 0.5$ m). When the sonic temperature was near the center of the chamber, we calibrated the temperature derived from these two measurements against a nearby RTD. For the calibration, the mean temperature derived from the sonic temperature sensor and the LiCor was calculated when in the middle position of the traverse. That derived temperature was then adjusted to equal the mean temperature measured by RTD-10 in Fig. 1.

## 2.3  Experimental Strategy

In order to determine the behavior of the LSC we followed the method of Brown and Ahlers (2006) and Xi et al. (2009). We use a ring of eight RTDs near the wall of the cylindrical chamber on the horizontal mid-plane of the chamber, and define the angular position ($\Theta$) of the RTDs as the total angular distance (clockwise) away from an arbitrary position. The temperature measured by any of the eight RTDs is then

$$T(\Theta) = \overline{T} + \delta cos(\Theta - \phi) \tag{3}$$

where $\overline{T}$ is the mean temperature in the ring, $\delta$ is the amplitude of the temperature variation among the eight RTDs and $\phi$ is the phase of the temperature variation. $\delta$ and $\phi$ are derived by fitting the measurements to Eq.(3). Notably $\phi$ represents the angular location of the updraft relative to the reference position.

During dry convection we applied this procedure independently on two different sets of eight RTDs, to characterize the behavior of the LSC and temperature distributions within the bulk. The placement of the two sets of RTDs formed an outer ring (Wall) and inner ring (Bulk) which were located 1 cm and 30 cm from the sidewall of the chamber.

We can describe the characteristics of the temperature in the chamber using only RTDs. If this were our only objective we would only need to run the chamber in dry conditions. However, in studying cloud properties we also need to describe the distribution of water vapor and by extension the saturation ratio. The traverse was introduced for the moist convection experiments in order to characterize the water vapor and saturation ratio at different locations in the flow. Due to physical limitations the center ring of RTDs cannot be used in tandem with the traverse.

Because our measurement of the water vapor mixing ratio is over a 12.5 cm path, we need to know how a volume/path averaged measurement will compare to an ideal (*i.e.* instantaneous, point) measurement. Because we cannot perform such

a measurement for the water vapor concentration, we used a large-eddy simulation (LES) to understand the effects of path averaging on water vapor concentration and temperature.

## 2.4 LES Results for Path Averaging

Our LES is the System for Atmospheric Modeling or SAM (Khairoutdinov and Randall, 2003), which has been adapted and scaled to the conditions of our chamber. In Thomas et al. (2019) the turbulent dynamics (energy dissipation rates, TKE and large scale oscillations) from the simulations have been matched with experimental values. The simulations reported here represent a $2\times2\times1$ m cell with $\Gamma = 2$ (*i.e.* the chamber without the cylindrical insert in place) with adiabatic side walls. The LES grid is $64\times64\times32$ with a spacing of 3.125 cm. The simulation was run with a time step of 0.02 s with a $\Delta T = 10$ K and $\overline{T} = 10$ °C. We exclude the first 50 min of simulation time from our results. The analyzed portion of the LES results span 70 min of simulation time (30 and 42 times greater than the period of the LSC respectively). We placed 41 virtual temperature and water vapor sensors in the center of the cell. The sensors are arranged in four lines of 11, centered at (1,1,0.5) (all distances in meters). Fig. 3 shows the locations.

We use a single grid box as an 'ideal' measurement. We simulated the sensor's path length by averaging the temperature and water vapor of $n$ adjacent points. We use the center bin as the reference and symmetrically average towards the ends of each line. The path length is then calculated by $d = n*3.125$ cm along the x and y axes. On the diagonal the path length is calculated by $d = n*3.125*\sqrt{2}$ cm. The path length for a single point, denoted as $d_0$, is the size of a grid box, 3.125 cm.

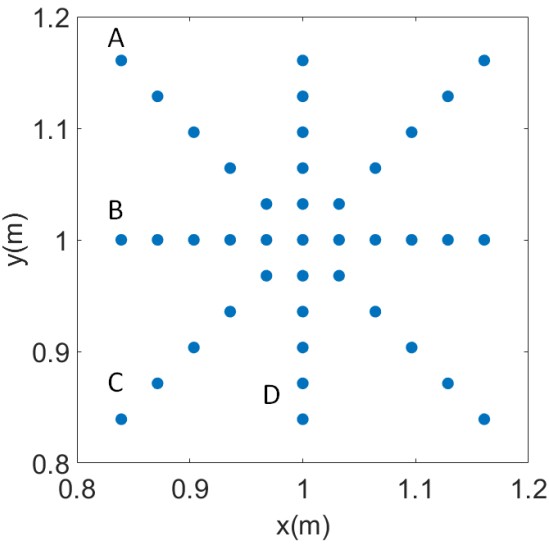

**Figure 3.** The positions of the virtual temperature sensors in the LES at $z = 0.5$ m.

The result of path averaging is shown in Fig. 4, which shows a plot of the standard deviation of temperature for a path length $d$, normalized by the standard deviation for $d_0$. Not surprisingly, as the path length increases, the normalized standard deviation of the measurement decreases. Also note that the curves from the four different lines of numerical sensors collapse. (The lines are denoted A, B, C, and D in Fig. 4 and Fig. 5). Note that although only results for $T$ are shown in Fig. 4, the data for $\sigma_r(d)\sigma_r^{-1}(d_0)$ and $\sigma_T(d)\sigma_T^{-1}(d_0)$ are identical due to the the non-dimensional units, and the same advective equations and diffusivity for both scalars. The LES results indicate that, over the path length of the LiCor and sonic temperature sensors, the standard deviations of $T$ and $r$ decrease by $\approx 8\%$. This result indicates that the measurements that we perform in the $\Pi$ Chamber do not capture the true variability in the system, but capture over $90\%$ of it.

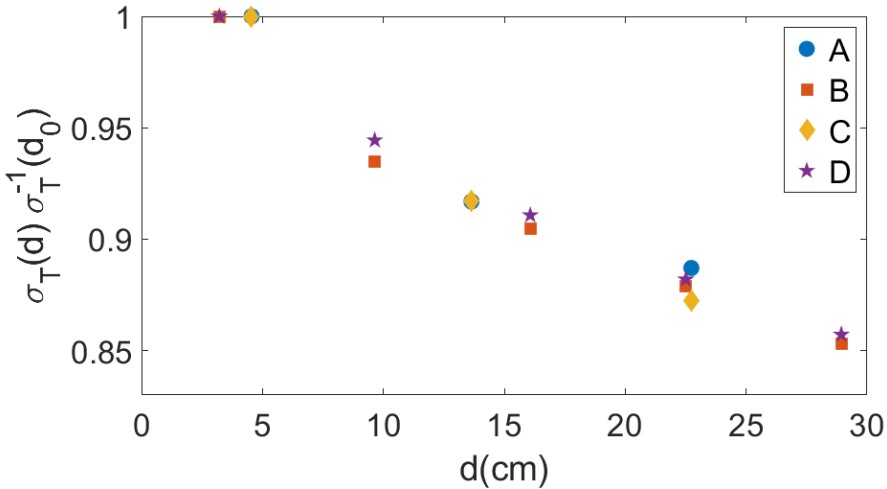

**Figure 4.** The normalized standard deviation of $T$ as a function of the sensors path length. Path averaging over $\approx 12.5$ cm results in a measured $\sigma(d)$ that is $\approx 92\%$ of $\sigma(d_0)$. See Fig. 3 for lines A, B, C and D.

The path averaged values for $r(d)$ and $T(d)$ were used with Eqn. 2 to calculate $S(d)$. In Fig. 5, $\sigma_S(d)\sigma_S^{-1}(d_0)$ is shown plotted against $d$. Over the same path length as the LiCor and sonic temperature sensors, $\sigma_S$ decreases by $\approx 19\%$. The percent decrease in $S$ over the path length of the LiCor is higher than $8\%$ due to the combined averaging of $r$ and $T$. We have shown that a path averaged measurement will underestimate the turbulent fluctuations. Path averaging is not the only type of averaging that is part of these measurement but we have determined that it is the most significant. For a further analysis of path averaging on the frequency spectra of $T$ and the analysis of time averaging see $Appendix\,A$.

## 3   Determination of Basic Characteristics of the LSC

We first ascertain the characteristics of the circulation, using only measurements of temperature (*i.e.* in dry conditions). For these conditions, we do not need to place the traverse with LiCor hygrometer and sonic temperature sensor in the chamber,

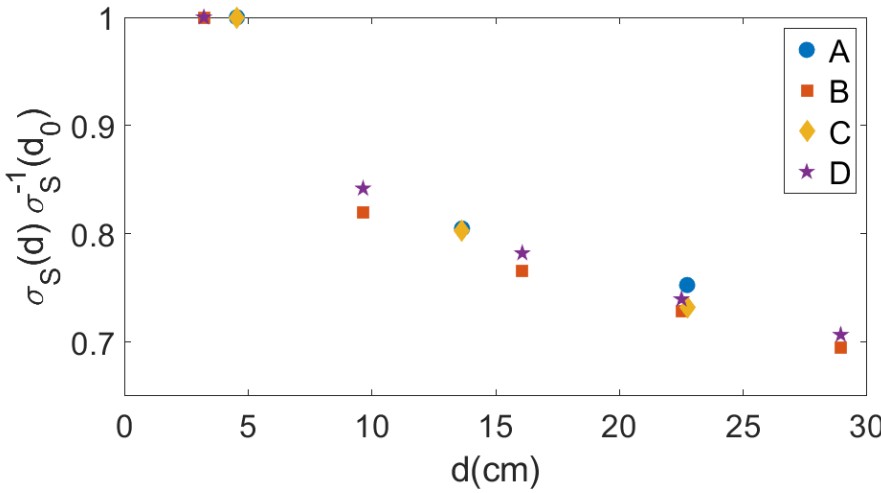

**Figure 5.** The path length averaged $\sigma_S(d)$ normalized by $\sigma_S(d_0)$ (the value measured in a single bin). Path averaging over $\approx 12.5$ cm decreases $\sigma_S$ to $\approx 81\% \sigma_S(d_0)$. See Fig. 3 for lines A, B, C and D.

so we can use the second ring of RTDs in the bulk of of the chamber (30 cm away from the side walls). In previous studies of Rayleigh-Bénard convection the first order moments of the circulation have been modeled as a single roll that spans the diameter of the cell using Eq. (3). This roll takes the form of a warm updraft along one side of the chamber with the cooler downdraft located along the opposite side. Due to the positive correlation between the vertical velocity and temperature, either variable can be used to find where the mean updraft is located (Shang et al., 2003). The location of the updraft is then used to determine the orientation of the circulation.

An example of the instantaneous temperature measured on the wall and bulk rings is shown in Fig. 6. In the figure, the temperature fluctuation, $T' = T - \overline{T}$ is shown against $\Theta$, where $\overline{T}$ is the mean temperature, averaged across all sensors in the ring and $T$ is the temperature measured by a single sensor at time $t$. The solid line is the least squares fit to the temperature measurements using Eq. (3). In both rings of RTDs, a sinusoid is a reasonable fit. The amplitude, $\delta$, and the orientation, $\phi$, were calculated from the fit; Fig. 7 shows the orientation of the circulation along the wall, $\phi_{Wall}(t)$, and in the bulk, $\phi_{Bulk}(t)$, as a time series. The difference between $\phi_{Wall}$ and $\phi_{Bulk}$ is smaller than the uncertainty in the fit. The uncertainty of the fit is $\pm 0.3$ radians and $\pm 0.1$ K for the orientation and the amplitude respectively. Over the course of our measurements the mean orientation for both precesses by $\approx 0.3$ radians. Both rings of RTDs show azimuthal oscillations of $\approx 0.6$ radians, which is inherent to the LSC (Brown and Ahlers, 2007b). The time series also shows that $\phi_{Wall}$ and $\phi_{Bulk}$ oscillate in phase.

In Fig. 8, the time series for $\delta_{Wall}$ and $\delta_{Bulk}$ are shown. The LSC does not show any cessations, with $\delta$ never approaching zero. The amplitude along the wall is consistently higher than the amplitude in the bulk. $\delta$ for both rings fluctuates around the mean by $\approx 0.1$ C. The amplitude of the LSC being highest near the wall is consistent with the circulation predominately following the walls of the cell (Qiu and Tong, 2001). The strength of the LSC decreases towards the center.

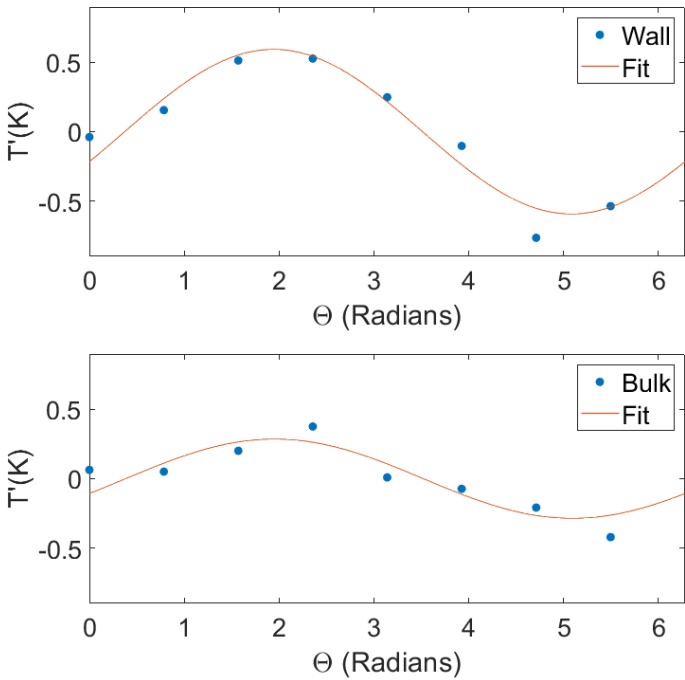

**Figure 6.** An example of the sinusoidal fit to the temperature fluctuations ($T'$) with respect to the azimuthal position of the RTDs ($\Theta$) along the wall (top) and in the bulk (bottom). The uncertainty in the temperature is too small to be seen on the graph. The goodness of the fit does not change over time.

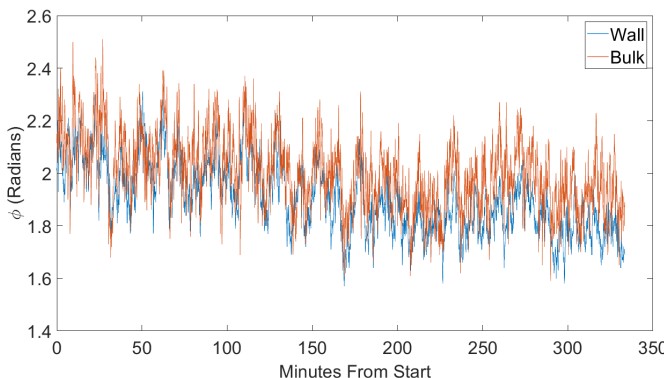

**Figure 7.** The azimuthal orientation ($\phi$) of the Large-Scale circulation for a $\Delta T = 12$ K along the wall (red) and 30 cm towards the center (blue). $\phi$ is essentially the same for both rings of temperature sensors.

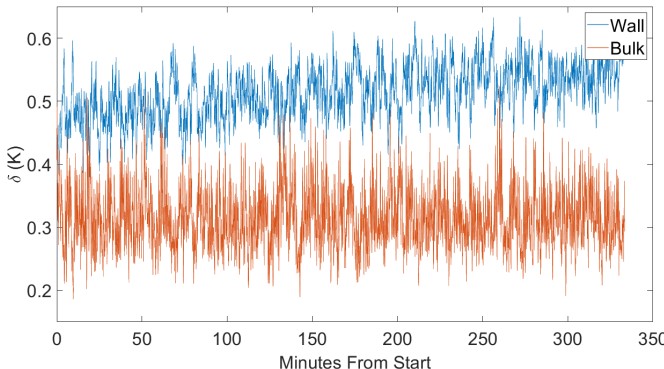

**Figure 8.** The amplitude ($\delta$) of the Large-Scale circulation for a $\Delta T = 12$ K along the wall (red) and 30 cm towards the center (blue). The amplitude near the wall is consistently higher than in the bulk.

As $\Delta T$ increases, the amplitude of the temperature on both rings increases, as shown in Fig. 9. For each $\Delta T$, $\overline{\delta_{Wall}} > \overline{\delta_{Bulk}}$. As $\Delta T$ increases, the standard deviation of $\delta$ ($\sigma_\delta$) also increases, showing that the variability of the LSC depends on $\Delta T$. Over the range of $\Delta T$s we have investigated, the amplitude for both rings increases linearly. In previous studies $f_0$ was shown to depend on $\Delta T$ (Qiu et al., 2004; Xi et al., 2006; Niedermeier et al., 2018) . Our periods we have measured in this study are essentially the same as those in Niedermeier et al. (2018).

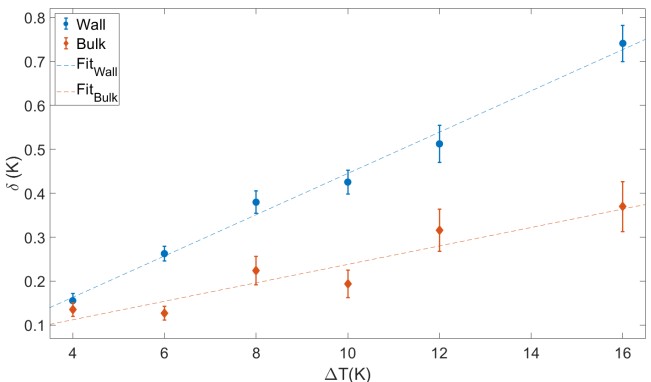

**Figure 9.** The amplitude, $\delta$, of the Large-Scale circulation along the wall (red) and 30 cm towards the center (blue) for a range of $\Delta T$s. The dashed lines are the linear fits to the wall(y=0.047$\Delta T$-0.25) and the bulk(y=0.021$\Delta T$+0.028). The uncertainties correspond to $\pm\sigma_\delta(\Delta T)$. At each $\Delta T$, $\delta$ is higher along the wall than in the bulk.

Our data show that the effects of the circulation are felt well into the bulk of the chamber, though, as expected, the amplitude of the circulation decreases towards the center. Our results also indicate that the circulation in the $\Pi$ Chamber has pronounced

azimuthal oscillations about a preferred orientation. The preferred orientation is a result of asymmetries and the instrumentation in the chamber. We will now address the impact of the LSC on the temperature distributions in the bulk of the chamber, using the RTDs in the bulk ring.

To minimize the effect of the chamber's temperature controls, which can fluctuate on the order of ten minutes, a high-pass Fourier filter (ifilter with a center of 3.3145 and width of 0.82846) with a cutoff of around five minutes was applied to the individual RTDs in the inner ring. The cutoff at five minutes is roughly four times larger than the period of the large scale circulation ($1/f_0$), where $f_0$ is the large scale circulation frequency. The angular deviation from the updraft was calculated from $\phi_{wall}$ using 16 bins of size $\pi/8$ radians to minimize the effect of azimuthal oscillations of the circulation. It should be noted that the azimuthal oscillations cause measurements from multiple RTDs contribute to the values calculated in each bin. With this done, the normalized standard deviation ($\sigma_T/\Delta T$) is shown in Fig. 10. Near the updraft ($\Theta - \phi_{wall} = 0$), $\sigma_T/\Delta T$ is lower than the rest of the chamber. The downdraft ($\Theta - \phi_{wall} = \pi$ and $\Theta - \phi_{wall} = -\pi$) curiously has a normalized standard deviation that is about twice the $\sigma_T/\Delta T$ in the updraft. In an ideal chamber $\sigma_T$ is likely the same for both the updraft and downdraft. The $\Pi$-chamber has several factors (for example viewing windows) that cause $\sigma_T/\Delta T$ to deviate away from the ideal case.

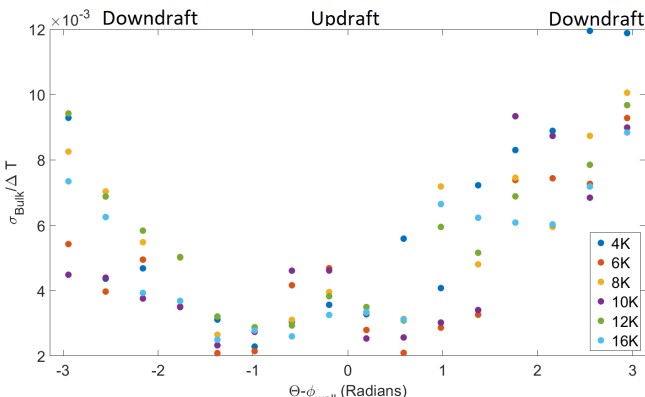

**Figure 10.** $\sigma_T$ normalized by $\Delta T$ as a function of the angular displacement from the updraft. The temperature was filtered using a high pass Fourier filter with the cutoff at around 5 min.

In Fig. 11 the skewness of $T$ is shown with the angular distance from the updraft of the LSC. The skewness is defined as $\mu_3 = \overline{(T - \overline{T})^3}/\sigma_T^3$. For each $\Delta T$ the highest skewness is measured when the temperature is near the updraft ($| \Theta - \phi_{wall} |= 0$) and lowest in the downdraft ($| \Theta - \phi_{wall} |= \pi$). Perpendicular to the axis of the circulation the distributions become symmetric. $\Delta T$ does not change the value of the skewness due to the normalization of the skewness. Multiple points in the same location are due to absolute value applied to $\Theta - \phi_{Wall}$. The same distance to the left and right of the circulation would then be expressed as two points at the same location away from the updraft.

The skewness is impacted greatly by rare events which likely contributes to the spread in values at each position from the updraft. In RBC rare events take the form of plumes that come from either the top or bottom boundaries. The positive skewness in the updraft is a result of warm plumes from the bottom surface. Alternately, cold plumes are more likely to pass through the downdraft, causing a negative skewness. Ideally in positions perpendicular to the LSC we expect warm and cold plumes to pass a sensor at an equal rate resulting in zero skewness. In Fig. 11, the spread of values perpendicular to the LSC ($|\Theta - \phi_{wall}| = \pi/2$) is likely due to the uncertainty in $\phi_{wall}$. Within the uncertainty, the off-axis region could be slightly closer to the updraft or downdraft. For example at $\Theta - \phi_{wall} = \pi/2$ warm plumes could be more frequent than cold plumes, despite being calculated at a spot were they should have equal probability. Directly across from those sensors ($\Theta - \phi_{wall} = -\pi/2$), the cold plumes may be more frequent than warm plumes. Both measurements would be represented at $|\Theta - \phi_{wall}| = \pi/2$ but the skewness would have the opposite sign.

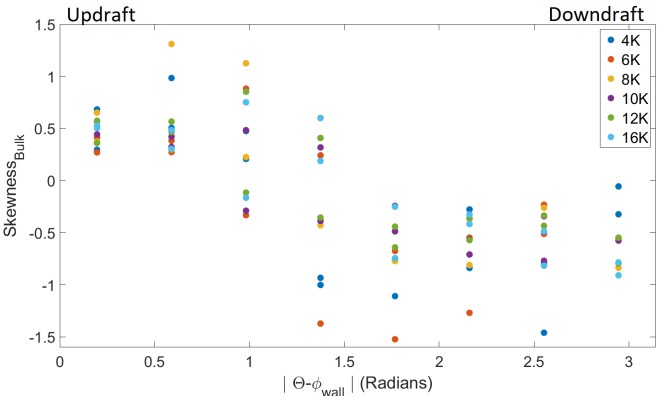

**Figure 11.** The skewness of the temperature measurements as a function of the angular distance from the updraft. The temperature was filtered using a high pass Fourier filter with the cut off at around 5 min. Near the updraft the temperature fluctuations are positively skewed. Near the downdraft the temperature fluctuations are negatively skewed. $\Delta T$ does not change the value of the skewness.

## 4   Moist Convection Results

Having established the basic characteristics of the large scale circulation, using measurements of the temperature, we turn to the scenario in which a difference in temperature and water vapor concentration between the top and bottom plates drives a convective flux of two scalars. The Rayleigh number is dominated by the temperature difference; the difference in water vapor concentration is small in comparison, which follows from Eq. (1). Given that, the behavior of the temperature field in the bulk of the chamber will be comparable in moist and dry conditions.

As noted in Sec 2, the inner ring of RTDs was removed to enable measurement of temperature and water vapor with the sonic temperature sensor and LiCor respectively. These instruments were mounted such that they were probing roughly the same volume; additionally the sensors were mounted such that they could be moved across the chamber on a traverse. The time

series of $r$, $T$ and $S$ are shown in Fig. 12 for $\Delta T = 12$ K. The sensors were near the updraft and downdraft for 4 hrs each. The sensors were in the center for 8 hrs. The figure clearly shows that the variance of the scalars is a function of the position in the large scale circulation. Near the downdraft of the circulation, the variance is the highest. This is consistent with the standard deviations near the downdraft shown in Fig. 10.

A Fourier analysis of $r$ and $T$ in the center of the chamber are shown on the right side of Fig. 12. The main peak is at a period of $\approx 72$ seconds which is due to the large scale circulation frequency, $f_0$, which has been shown to be caused by the azimuthal oscillations of the LSC (Xi et al., 2009). Harmonics of $f_0$ can be seen in both measurements. For a more detailed analysis of Fourier spectra in the chamber, see Niedermeier et al. (2018).

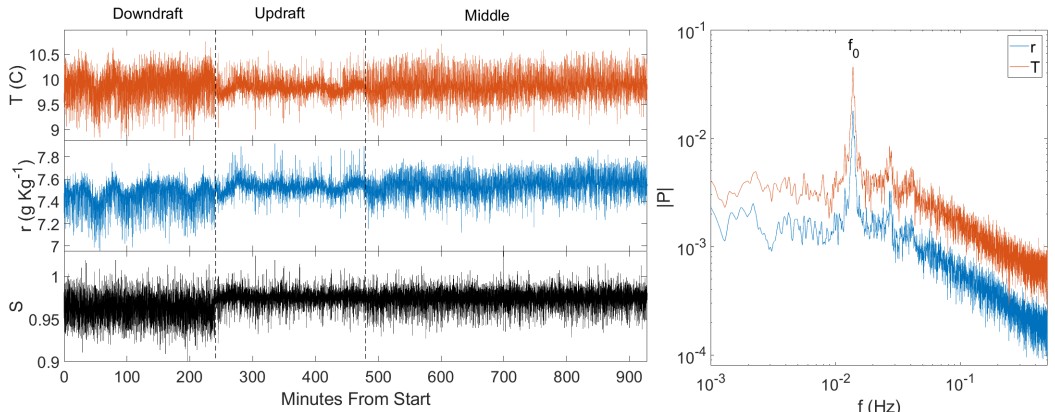

**Figure 12.** The left panel shows the time series of temperature ($T$, red), water vapor mixing ratio ($r$, blue) and the saturation ratio ($S$, black) at different positions in the $\Pi$-chamber. The time series only include periods in time where the chamber is in steady state conditions. At the beginning of the time series the traverse was near the downdraft side of the chamber. At 240 minutes from the start the sensors were moved to near the updraft. At 480 minutes they were moved to the center of the chamber and remained there until the end. The right panel contains the Fourier spectrum of $r$ and $T$ while the sensors are in the center of the chamber. The oscillation frequency, $f_0$, corresponds to a period of $\approx 72$ seconds. The spectra are smoothed for clarity.

As another perspective on these measurements, we show the probability distribution functions (PDFs) for $T'$ in the top of Fig. 13. (It should be noted the fluctuations in the top and middle of Fig. 13 are in relation to the temporal average at each individual measurement position along the traverse.) The standard deviations and skewness of $T$, $r$ and $S$ are presented in Table 1. The standard deviations of $T$ in the moist case are consistent with the trends seen in Figs 10 and 12 where the downdraft of the circulation has the highest variance. The skewness for $T$ is positive near the updraft, and negative near the downdraft. These values for the skewness are consistent with the values calculated from the RTDs in Fig 11. The middle of Fig. 13 shows the PDF of $r'$; the overall shape of the distributions is similar to the distributions of $T'$. Like $\sigma_T$, $\sigma_r$ is lowest near the updraft and highest near the downdraft. The skewness for $r$ follows the same trend as $T$ with a positive skewness near the updraft and a negative skewness near the downdraft. Taken together, the distributions of $T'$ and $r'$ reinforce the phenomenological picture

that warm, humid plumes are more likely to be seen in the updraft region of the chamber. The opposite is true for the downdraft, where the presence of cold, low $r$ plumes lead to both distributions being negatively skewed.

| Position | $\overline{T}$(C) | $\sigma_T$(K) | $skewness_T$ | $\overline{r}$(g/kg) | $\sigma_r$(g/kg) | $skewness_r$ | $\overline{S}$ | $\sigma_S$ (%) | $skewness_S$ |
|---|---|---|---|---|---|---|---|---|---|
| Updraft | 9.83 | 0.12 | 0.50 | 7.53 | 0.05 | 0.50 | 0.97 | 0.69 | -0.50 |
| Middle | 9.88 | 0.19 | 0.26 | 7.55 | 0.08 | -0.21 | 0.97 | 0.82 | -0.64 |
| Downdraft | 9.89 | 0.22 | -0.52 | 7.45 | 0.09 | -1.08 | 0.96 | 1.19 | -0.18 |

**Table 1.** The statistics for $r$, $T$, and $S$ at each position along the traverse during moist conditions and $\Delta T$=12K.

In the bottom of Fig. 13 is a plot of the probability distributions of $S-\overline{S}_{middle}$. We have subtracted the mean of the saturation ratio from the middle of the chamber to highlight the fact that the downdraft has a lower mean than does the updraft or core region of the chamber. The distributions of $S-\overline{S}_{middle}$ are quite similar in the updraft and in the middle of the chamber, while the distribution in the downdraft is broader (see Table 1).

Because of the correlation between temperature and water vapor concentration in the chamber, a change in either $r$ or $T$ will not *a priori* lead to a change in $S$. A positive fluctuation in $T$ could be associated with a positive fluctuation in $r$ such that the ratio of $r$ and $r_s$ do not change. (See Chandrakar et al. (2020a) for a more complete discussion of the correlation between $r$ and $T$ and the corresponding changes in $S$.) Our data show that the skewness of $r$ and $T$ are comparable in both sign and magnitude in the updraft region of the circulation. $S$ however, is negatively skewed at each location in the chamber.

## 5 Conclusions

The convection-cloud chamber at Michigan Tech, the $\Pi$ Chamber, is a Rayleigh-Bénard convection cell, designed for studies of interactions between turbulence and cloud microphysics. Through measurements of the temperature in the chamber, we have shown that the large scale circulation is a single roll with a fixed overall orientation, but with pronounced oscillations about the mean position, typical of the large scale circulation in Rayleigh-Bénard convection.

To determine the saturation ratio in the chamber, we measure water vapor concentration and temperature, simultaneously, to get the saturation ratio, $S$. Because point measurements of water vapor concentration are not currently possible, we have verified that our path averaged measurements capture an acceptable fraction of the true variance in the system, using a combination of measurements and large eddy simulations. The LES shows that $\sigma_T$ and $\sigma_r$ decrease by $\approx 8\%$ from their true values when the measurement is averaged over approximately 12 cm, as ours are. The corresponding decrease in $S$ is $\approx 19\%$. (Path averaging is more pronounced for $\sigma_S$ due to the combined averaging from $r$ and $T$.) The LES shows that path averaged measurements do underestimate but still represent a sizable portion of the turbulent fluctuations of $r$, $T$, and $S$.

We show that water vapor concentration and temperature distributions in the updraft and downdraft are qualitatively similar. For example, both scalars in the updraft are positively skewed and have a higher mean than the center. Combining these measurements into $S$ shows turbulent fluctuations that are caused by fluctuations in $r$ and $T$. $S$ is consistently negatively skewed even in the updraft where both $r$ and $T$ are positively skewed. In the downdraft the distribution of $S$ is more negatively

skewed than the updraft. While our results show significant fluctuations in $r$, $T$ and $S$, the true variability on scales felt by cloud droplets would likely be higher than what we have reported because of the path lengths of our sensors.

As noted in the Introduction, one of the primary motivations to understand the spatial and temporal variability of the saturation ratio in the chamber is to then relate it cloud droplet growth. In previous analyses of microphysics in the chamber, zero and first order models of the variability in the saturation ratio have been used (Chandrakar et al., 2016, 2020a). The results presented here indicate that while these models capture the essential variability (standard deviations) of $T$, $r$, and $S$ in the center of the chamber, spatial variations of the mean in the chamber may affect, for example, where droplets preferentially activate or evaporate. How these spatial differences impact cloud droplet distributions and how cloud droplets alter the saturation field are ongoing topics for both experimental and modeling efforts.

Future work will focus on how the fluctuations of $S$ change upon the transition from moist to cloudy conditions. The in-cloud saturation field is dependent on the initial $S$ (the moist conditions that we have shown in this paper) and the influence of cloud droplets. The presence of droplets is expected to buffer the fluctuations in $S$, but the magnitude is currently unknown.

**Appendix A**

We have shown that a path averaged measurement will underestimate the turbulent fluctuations. This type of averaging likely acts as a low pass filter, with the high frequency fluctuations being removed. In Fig. $A1$ the power spectra of $T$ are shown for several different path lengths. As the path length is increased, the higher frequencies are proportionally removed at a faster rate, decreasing the slope of the spectra. Over the path length of the LiCor ($\approx 12.5$ cm) the spectra are noticeably impacted by the path averaging, but represent the overall shape and magnitude of the spectra of temperature reported by the single bin.

Not only does the path length of a sensor artificially dampen scalar fluctuations, but the sensor's time averaging must have a similar affect. For temperature, one cause of time averaging is the sensor's thermal mass. Ideally a thermometer would have a small mass, allowing it to rapidly respond to changes in temperature. This is one of the reasons that we have used both rtds (which have a non-negligible thermal mass) and the sonic temperature sensor, which does not. The other cause is digital averaging over multiple samples from the same sensor. In our case, the high speed temperature system digitally averages over one second to output data at 1 Hz. This section will discuss the impact of digital time averaging on the fluctuations of $T$ and $S$.

To estimate the effect, we took the output of the virtual sensor of the LES in each of the four corners and the center (see Fig. 3). The averaging time ($t*$) was simulated by applying a moving average of varying size to the LES output. In Fig. $A2$ the standard deviation was then calculated for the averaged time series and normalized by the standard deviation of the original time series ($\sigma_T(t_0)$). Over the averaging time of the sonic temperature sensor (one second), the standard deviation decreases to $\approx 94\%$ $\sigma_T(t_0)$. This percentage is slightly higher but still comparable to the 12.5 cm path averaged measurement.

In Fig. $A3$ the time averaging for $S$ was calculated using the time averaged values of both $r$ and $T$. When the value was averaged over 1 second, the fluctuations in $S$ decrease to $\approx 87\%$ of the ideal value. Notably, the decrease in fluctuations of $S$ for the path averaged measurements showed a decrease to $\approx 81\%$ $\sigma_S(t_0)$. Clearly time averaging of the sensors should not be ignored; however in our case the path averaging of the sensors is the main contributor to suppression of fluctuations.

*Data availability.* Data is available through Digital Commons at Michigan Tech, https://doi.org/10.37099/mtu.dc.all-datasets/3 .

*Author contributions.* JA, ST, and PP conceived and carried out the experiments, analyzed the data, and wrote the manuscript. RS and WC conceived the experiments, participated in analysis and discussion, and wrote the manuscript.

*Competing interests.* The authors declare no competing interests.

*Acknowledgements.* We thank the US National Science Foundation (AGS-1754244) and the US Department of Energy (DE-SC18931)
for funding. Superior, a high-performance computing infrastructure at Michigan Technological University, was used in obtaining results
presented in this publication.

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

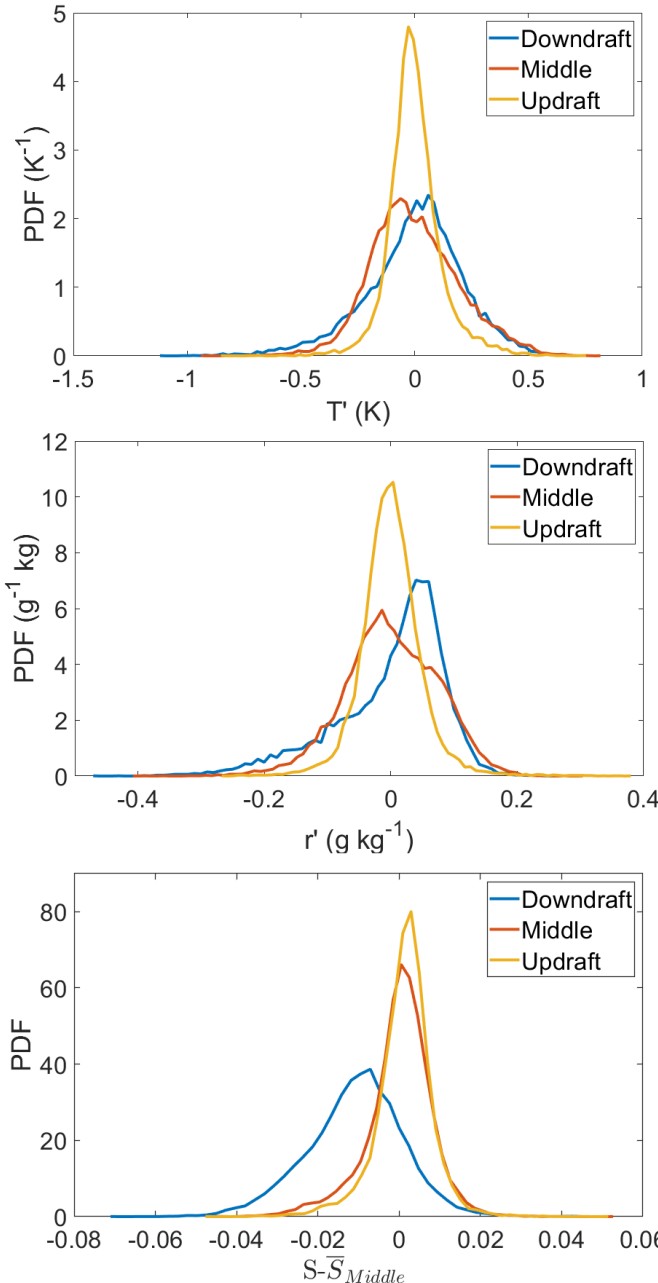

**Figure 13.** The probability distributions of $T'$(top), $r'$(middle), and $S - \overline{S}_{middle}$ (bottom) near the updraft, downdraft and middle. For each region a high pass filter was applied to $r$ and $T$ with a cut off of $\approx 5$ min, to remove low frequency oscillations due to the slight drift in the chamber controls. We have plotted the distributions $S - \overline{S}_{middle}$, not $S'$ to highlight the fact that the downdraft has a lower mean relative the middle and updraft.

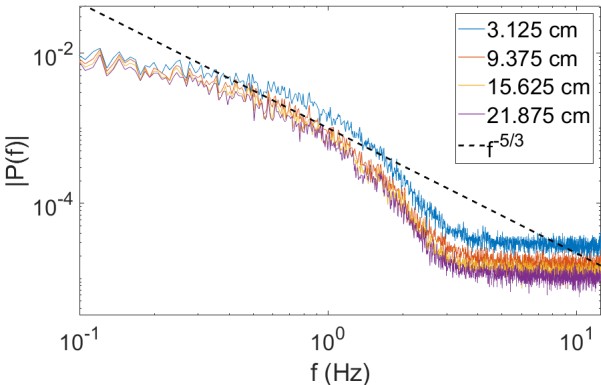

**Figure A1.** The spectra of the temperature measurement for several different path lengths. These spectra are averaged along line B and are smoothed for clarity. The dashed line is a power law ($f^{-5/3}$) included as a reference.

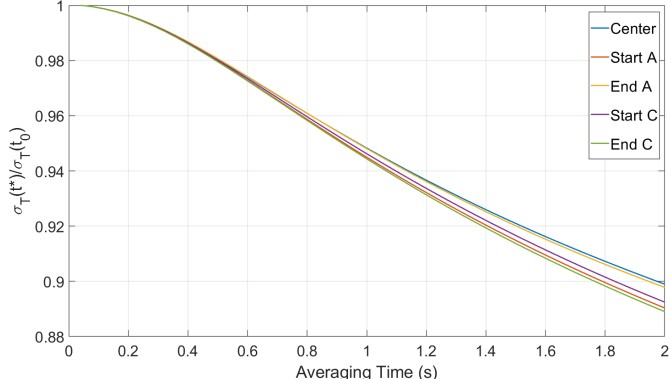

**Figure A2.** The time averaged $\sigma_T(t*)$ normalized by $\sigma_T(t_0)$ (the standard deviation of the raw temperature time series). Over the time averaging of the sonic temperature sensor, $\sigma_T$ decreases to $\approx 94\%$ $\sigma_T(t_0)$.

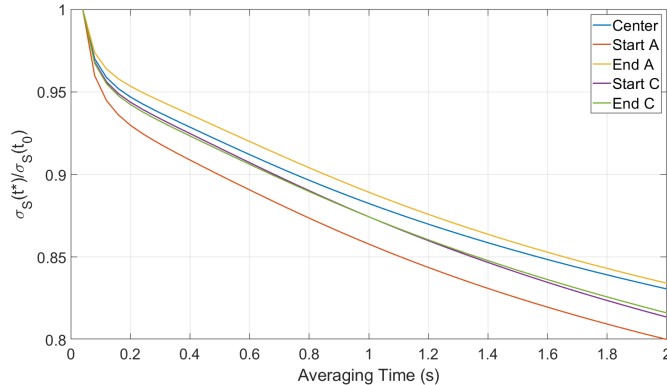

**Figure A3.** The time averaged $\sigma_S(t*)$ normalized by $\sigma_S(t_0)$ (the standard deviation of the raw saturation ratio time series). Over the time averaging of the sonic temperature sensor, $\sigma_S$ decreases to $\approx 87\%$ $\sigma_S(t_0)$.