# Peer review of "Effects of the Large-Scale Circulation on Temperature and Water Vapor Distributions in the Π Chamber"

_Atmospheric Measurement Techniques, 2021_

## Author Comment (AC1)

**Response to Reviewers**

We appreciate the thorough consideration of the manuscript from both reviewers. We have incorporated the comments into the manuscript without exception. We are gratified that both reviewers found the analysis and level of detail presented in the manuscript as "clear" and "very good".

Our responses below are shown with the specific comment from the reviewer in red and our response directly below in blue.

**Referee 1**
**General comments:**
The authors investigate the structure of large-scale circulations (LSCs) in turbulent Rayleigh-Benard convective with aspect ratio 2, and in the presence of multiple scalars (temperature, water vapor mixing ratio, and saturation ratio). This study is motivated by the Michigan Tech Pi chamber, a unique facility to study interactions between aerosols, turbulence, and cloud microphysics. While the Pi chamber has yielded many valuable insights, the effects of spatial heteorogeneity in the chamber on the scalar fields has not been investigated in detail. The authors present clear evidence of LSCs in the chamber, which have a non-negligible influence on the skewness of temperature, water vapor mixing ratio, and saturation ratio. The manuscript is clear, well-written, and of an appropriate focus and scope for AMT. I have some specific comments below I recommend that the authors address, but these should not be a barrier to publication.

**Specific comments:**
1. r is defined as both droplet radius in l. 23 and water vapor mixing ratio in l. 53. I would suggest a change in notation.

- We agree with the reviewer. We have changed r so that it solely defines the water vapor mixing ratio. The droplet radius is now defined as R.

2. The saturation ratio is used several times (ll. 22, 35, 82, 92) before it is actually defined in Equation 2; I would suggest defining it the first time it is used.

- We have added the definition in the second paragraph.

3. l. 103 What approximation of the Clausius-Clapeyron equation is used to calculate saturation vapor pressure? I'm guessing that it is probably based on an empirical fit that accounts for the variation of the latent heat of vaporization with temperature, but more detail would be beneficial.

- We have used the Magnus equation to calculate the saturation vapor pressure. This approximation accounts for the variation of the latent heat of vaporization with temperature. We have added this point to the manuscript.

4. Fig. 2 caption "temperature calibration of the sonic temperature" sounds redundant; consider rewriting.

- The caption of Fig. 2 has been changed to now say "calibration of the sonic temperature".

5.  l. 146 "The grid spacing is 3.125 cm" in reference to the LES. Is this only for the horizontal grid spacing, or also the vertical grid spacing? So the LES grid is 64x64x32 points? This seems very coarse to me, and it doesn't appear that the authors are using bin microphysics (which has a significant computational cost)? Why not run at a higher resolution? Have grid convergence tests been done to demonstrate that this resolution captures turbulence satisfactorily within the Pi chamber?

- The LES grid is 64x64x32 points.  Both the horizontal and vertical grid spacing of 3.125cm. The LES grid spacing is chosen such that the spatial scale is at least 30 times the Kolmogorov length scale - and lie in the inertial range according to Monin and Yaglom (2013). The turbulent dynamics - energy dissipation rates, TKE and large scale oscillations - from the simulations have been matched with the experimental values (Thomas et al. JAMES 2019). We have added this clarification to the manuscript where the LES is introduced.

6.  l. 147-150 Discussion of 50 min spinup and 70 minutes of data analyzed from the LES. It is helpful to have these listed in dimensional values, but from a fluid mechanics perspective what matters is the number of independent samples (or integral timescales) that one is spinning up and averaging over. Can these values also be reported in the text?

- The 50 min spinup and 70 min of data are 30 and 42 times greater than the period of the LSC respectively. We have added this to Section 2.4.

7. l. 201 High pass filter "with a cutoff of around 5 minutes" Is it possible to give the exact temporal filter width here, rather than giving an approximation? Also, what is the filter kernel that is used here (e.g. is iit a spectral cutoff filter, Gaussian, box filter, or something else...)?

- The filter used is a spectral cutoff filter with a center of 3.3145 and width of 0.82846. We have included the width and center values in line 201.

-8. l. 202 Discussion of the period of large-scale circulations. The authors presented evidence that the amplitude of the LSC varies with the temperature difference between the top and bottom walls, but what about the frequency of the oscillation? I'm assuming this is discussed in the literature, so it would be useful to highlight previous studies here.

- The frequency of the LSC is dependent on the temperature difference between the top and bottom plates of the chamber (See Niedermeier et al., 2018). We have clarified this in the manuscript in Section 3 and we have added additional references to support the finding.

**Technical corrections:**
None needed, to my knowledge.
**Citation**: https://doi.org/10.5194/amt-2021-13-RC1

---

## Author Comment (AC2)

**Response to Reviewers**

We appreciate the thorough consideration of the manuscript from both reviewers. We have incorporated the comments into the manuscript without exception. We are gratified that both reviewers found the analysis and level of detail presented in the manuscript as "clear" and "very good".

Our responses below are shown with the specific comment from the reviewer in red and our response directly below in blue.

**Referee 2**
**Overview**
The manuscript presents a description and analysis of the laboratory experiments performed in order to characterize the thermodynamic properties of the large-scale circulation formed within the turbulent Rayleigh-Benard convection in the Π Chamber. This characterization is based on the measurements of temperature, water vapor and saturation ratio at a number of points in the mid-plane of the chamber under dry and moist conditions. Although the obtained results cannot be directly related to physical processes in the atmosphere, the true relevance of this work is that it provides a valuable context for a few previous laboratory studies at the Π Chamber which focused on cloud droplet activation and condensational growth in a turbulent flow. The experiments are described in detail and the quality of the analysis is very good. I recommend publication after addressing the comments below.

**General comments**
In my opinion, the great advantage of the paper is the experimental work done as well as the detailed description of the experiments and the following analysis. As far as I understand, the objective and the motivation for the study was "to measure the spatial and temporal variability in r, T and S in order to determine how closely the models of reduced dimensionality capture the true variability in the chamber" (line 82). The first aspect was addressed very well, the second only shortly mentioned in the conclusions (line 282).

You concluded that "models capture the essential variability of T, r, and S in the chamber". Please be more specific about what the models capture. Do you mean the single roll LSC pattern or the PDFs of the fluctuations? I presume that the issue is subject to ongoing work, however I suggest discussing briefly some implications of your results in the last section, e.g. on the interpretation of the previous findings from the Pi chamber.

- We agree with this point. When we mention the essential variability of T, r, and S we are referencing the PDFs of the scalars. The models capture the standard deviations well but cannot account for the spatial differences in the mean or skewness. Part of our ongoing research will be towards how these spatial differences impact cloud droplet distributions and how cloud droplets alter the saturation field. We have included this point in the second to last paragraph of the Conclusions.

I suggest collecting all the information about the instrumental configuration and the execution of the experiments in the methods section. In a few places, I wanted to request additional details but then found them further in the paper (in sec. 3 or 4). For instance: what was the position of the sonic with respect to the hygrometer and whether the paths were coincident (given in line 231), why the inner ring was not used in the moist experiment (given in line 230), how many and what runs were performed (deduced from the figures), how long were the measurement series (given in the figures and their

captions).

For the convenience of readers, the strategy of the study may be explained in a straightforward way in one additional paragraph. It became evident to me only after reading the result sections that: the role of the outer ring was to track the orientation of the LSC, the role of the inner ring was to characterize the temperature field in the dry experiment, the traverse was (probably) introduced at the expense of the inner ring to characterize the humidity and saturation fields in the moist experiment, and the crucial assumption is that the behavior of the temperature field does not change significantly between dry and moist conditions (line 228) which can be justified by comparing the terms of Eq. (1). You can consider organizing sec. 2 into subsections about the facility, instrumentation, experimental strategy and influence of path averaging.

- We have implemented the reviewer's suggestion to clarify our approach in section 2.

**Specific comments**
1. You seem to use "we" and "our" in two meanings: the authors of the paper as well as the broader research group (e.g. line 6, line 34, line 76). Please avoid such a confusion because it may lead to the misunderstanding of what you did within your present study.

    - We agree with the reviewer that usage of "we" and "our" was confusing. We have tried to revise the manuscript to remove the confusion. In the revised version of the manuscript, "we" and "our" are used to signify the current authors and the current investigation.

2. Please determine whether r denotes the concentration of water vapor molecules ($cm^{-3}$), water vapor mixing ratio (per unit mass of dry air, g/kg) or water vapor mass fraction (specific humidity, per unit mass of moist air, g/kg), and use the same term consistently throughout the manuscript. The same symbol should not be used for droplet radius (line 23).

    - See the response to comment 1 from reviewer 1. The units of r are shown in the middle panel of Figure 12. We are using the mass of water vapor to mass of dry air.

3. Please mind the difference between saturation ratio and supersaturation. If S denotes the former, then the equation in line 23 should read dr/dt ~ (S-1)/r.

    - We appreciate the reviewer's attention to detail in this. The equation in line 23 has been changed to dR/dt ~ (S-1)/r so that S represents the saturation ratio.

4. Lines 58-73. In your thorough introduction you summarize previous studies on the LSC in turbulent RBC and describe the findings concerning mean flow patterns. Has anyone studied higher order moments (variance and skewness) as you did or is your work unique in this aspect?

    - Few papers have studied the higher order moments of RBC. We have added this point to the discussion in the Introduction.

5. Lines 94-98. You specified boundary conditions in terms of temperature. What are they in terms of humidity? Are the walls kept saturated in moist convection experiment?

- In the moist convection experiments the boundaries are kept at saturation. We have added a description of the moisture level of the boundaries to line 107.

6. Lines 116 and 119. You sampled r and T at 1 Hz with your hygrometer and sonic. Does this frequency correspond to the time constant of the instrument (i.e. the effective averaging time is 1 s) or to the rate at which you recorded the data (i.e. the averaging time is smaller but you save a result once in 1 s)? This question is relevant because both instruments are capable of much faster sampling.

- The LiCor hygrometer outputs at 1Hz but has an effective averaging time (response time) of 5Hz. The averaging time of the sonic temperature is 1Hz. We have included the digital averaging time in section 2.2.

7. Lines 119-120. "The sonic temperature sensor operates on the same physical principle as a sonic anemometer, using the Doppler shift in emitted and detected sound waves to derive the quantity of interest, using the known speed of sound." Are you sure the operation principle of your instrument is as described? As far as I know, most sonic anemometers rather than Doppler shift determine the transit times of an acoustic signal in two opposite directions out of which flow velocity and speed of sound can be estimated. The latter depends on air humidity. Hence, r can be inferred.

- We agree. The sonic temperature sensor measures the transit time of an acoustic signal. Using the transit time, the speed of sound is measured from which we are able to calculate the virtual temperature. We have amended this section.

8. Line 123. How did you calibrated the sonic thermometer - does the procedure involves any adjustments in the instrument itself or did you fit some calibration curve (e.g. straight line) to the recorded values?

- The sonic temperature sensor was calibrated to the mean temperature measured by the center RTD. We used this method because offsets in the sonic temperature sensor and the LiCor can cause an offset in the temperature when converting from the virtual temperature to the actual temperature.

9. Sec. 2.1. I appreciate the analysis of the influence of averaging along the measurement path on resulting turbulent fluctuations.
    1. In addition to spatial averaging, I suggest considering temporal averaging. Assuming, the time in your LES can be interpreted as physical time, the sampling of the "ideal measurement" is 50 Hz whereas the sampling of the real measurement is 1 Hz (see comment 6 first). Taking this into account, spatial averaging should have smaller influence and comparing the spectra (Fig. 6) turns out rather unnecessary.

    - We agree that the effects of time averaging on the measurements should not be ignored. To address this point we have included an analysis of temporal averaging and the LES results in Appendix A. For our system we show the time averaging does artificially remove high frequency fluctuations of T and S. In our specific case, the path lengths of the LiCor and sonic temperature cause the path averaging to have a larger impact than the time averaging. We have also moved Fig. 6 to Appendix A.

2. You analyzed the averaging effects only for the central part of the chamber and the paths which are symmetric with respect to the central point. However, in the moist experiment the hygrometer and the sonic seem to be located far from the center for updraft and downdraft (the exact position is not given). Can you explain the choice? The path averaging can possibly exert more influence on the measurements further from the center. I am aware that repeating the LES is rather not feasible but having the timeseries at your 41 points, I would suggest considering asymmetrical and off-axis paths. This will add more points to Fig. 4.

   - We chose the symmetric paths in the center of the chamber because the starting location (d_0) does change the curve of Fig. 4. For example, if we calculated the normalized standard deviation $(\sigma_T(d)/\sigma_T(d_0))$ starting on opposite ends of line A in Fig. 3 and averaging towards the center, we would be measuring from an asymmetric starting place but averaging along the same line. In this case both lines would start and finish at the same place, however both curves would have a slightly different shape. We found the starting location and direction of path averaging created variability in the curves of $(\sigma_T(d)/\sigma_T(d_0))$ and $(\sigma_S(d)/\sigma_S(d_0))$. We found the curves shown in Figs. 4 and 5 to best represent the average of the asymmetrical curves.

3. Grid boxes in a LES have finite volume. In fact, your single grid box have the volume significantly larger than the measurement volume of the hygrometer and the sonic given in sec. 2. This fact can be used as the argument in favor of your measurements resolving considerable portion of turbulent statistics.

   - We agree with this statement. We have emphasized this point in the conclusions section.

10. Lines 178-178. "Due to the positive correlation between the vertical velocity and temperature, either variable can be used to find where the mean updraft is located." Can you provide a reference which proves that?

   - The positive correlation is due to cool plumes falling and warm plumes rising in a convective environment. We have added a citation to support this.

11. Fig. 7. Please add error bars to the points. This should be straightforward as you performed a direct measurement of temperature.

   - The uncertainty in the temperature differences measured by the RTDs is ±0.001 K. Note that what is plotted in this figure is the temperature difference between a reference rtd and another one in the ring. We calibrated the rtds against each other. The uncertainty is based on the residual values we get once we apply the corrections. In Fig. 7 (now Fig. 6) this uncertainty is smaller than the size of the points. We have stated the uncertainty in Section 2.2.

12. Line 186. "smaller than the uncertainty in the fit". Please provide a typical value for the uncertainty of the fitted orientation and amplitude. Please comment whether the goodness of fit changes in time.

- The uncertainty of the fit is ±0.3 Radians and ±0.1 K for the orientation and the amplitude respectively. We have added these uncertainties to Section 3. The goodness of fit does not change with time.

13. Line 187. "azimuthal oscillations". Are they related to the drift in chamber controls which you filter later or are they the inherent property of the LSC? Does this oscillation have the frequency $f_0$ determined in sec. 4?

- The azimuthal oscillations are inherent to the LSC. We have included a reference to Brown and Ahlers (2007b) to this line to support this statement.

14. Line 195 and Fig. 10. Please fit the lines and draw them in the plot.

- We have included the linear fit lines to Fig 10.

15. Line 197. Do you mean the orientation of the LSC is the same for different experimental runs? Does the presence of the traverse in the moist experiment have any influence?

- The orientation of the LSC is the same for different experimental runs. The traverse does have an influence on the orientation of the LSC. The paper has been changed to include this statement.

16. Line 204. As far as I understand, after the binning of the individual instantaneous measurements you calculated the standard deviation within each bin separately. This implies that, in principle, the measurements from different sensors contribute to each resulting σ value. Please clarify.

- Different sensors contribute to the standard deviation of each bin. This has been clarified in the text.

17. Figs. 11, 12 and the relevant discussion. In my opinion, introducing absolute value of angular position versus updraft is misleading. I suggest using the deviation itself with its sign specifying the side of the chamber. In the figures, please use the entire range [-π,+π] or consider presenting the data in the form of polar plots. For convenience, you may add additional "updraft" and "downdraft" labels.

- Fig. 11 has been changed to express the standard deviations to the left and the right of the updraft in the chamber. For Fig. 12 we feel that the data is better represented with the absolute value of the angular deviation from the updraft because there is a high amount of symmetry to the left and right of the updraft.

18. Line 233. Did you perform the moist experiment for only one temperature setting because of the required duration? Is there any specific reason for choosing ΔT = 12 K? Please explain the strategy (see general comments).

- The moist convection experiments were performed for ΔT=4K to 16K. ΔT = 12 K was chosen because it is representative.

19. Line 233. What was the exact position of the sensors for updraft and downdraft regions in the moist experiment? Was it 30 cm from the center where the inner ring of thermometers was located in the dry experiment? Please provide those details in sec. 2 (see general comments).

- We agree and have included the distance of the sensors to the sidewall in section 2.2.

20. Line 238. Can the circulation frequency be attributed to azimuthal oscillations, sloshing mode or did you mean it is related to the LSC turnover time?

- The circulation frequency can be attributed to the azimuthal oscillations.

21. Figs. 14, 15 and the relevant discussion. Are the fluctuations T', r' calculated with respect to the time average for each measurement point or with respect to the global mean values in the chamber similarly to S in Fig. 16?

- The fluctuations are relative to the time average at each measurement point. We have added this to the paper.

22. Figs. 14, 15 and 16. For better readability, I suggest combing those three figures into one with 3 panels. You can refine x axis range to show the central part of the PDFs.

- We have combined the figure into three panels.

23. Lines 240-261. I suggest providing the results in a table: mean values, standard deviations, skewness of T, r, and S.

- We have reorganized the mean values, standard deviations, and skewness of r, T, and S into Table.1.

24. Line 274. "qualitatively similar". Do you mean their Gaussian-like shape or some other property? Please clarify.

- Qualitatively similar refers to the both r and T having similar a similar mean relative to the center and skewness at each position. The text has been changed to reflect this.

**Minor issues**

1. Line 12. "consistently" – do you mean at each position? Please clarify.

- Consistently refers to each measuring position. We have clarified that point in the abstract.

2. Line 60. "near one or two" – do you mean the whole range or two separate values?

- Near one or two refers to the range of aspect ratios between one and two.

3. Line 87. "respects" – do you mean "respects" or "aspects"?

- "Respects" has been changed to "aspects".

4.  Line 123. "sonic temperature" – do you mean "sonic thermometer"?

    - "Sonic temperature" was changed to "sonic temperature sensor".

5.  4 and 5. Please increase the size of the symbols.

    - We have increased the size of the symbols.

6.  Line 169. Wrong formatting of the unit "cm".

    - We have changed "cm" to be in the proper format.

7.  Line 202. "period of the LSC" is unclear and $f_0$ is not defined at this point. Please refer to sec. 4 and Fig. 13 here.

    - We have now defined $f_0$ in this line.

8.  Lines 218, 220, 221, 222. I assume you meant $\pi/2$ instead of $\pi$ to denote the directions perpendicular to the LSC.

    - Yes, we did mean $\pi/2$ instead of $\pi$. We have corrected the text.

9.  Referring to equations. The house standard of AMT is Eq. (n).

    - We have corrected the references to equations to the house standard.

10. Please add DOIs. This enables the reader to access the cited paper through a single click.

    - We have added DOIs to the citations.

**Citation**: https://doi.org/10.5194/amt-2021-13-RC2

---

## Author Response (AR2)

Dear Dr. Kanji,

We appreciate your efforts to expedite processing of the paper. We have addressed all of the points in your previous communication. See below for specific replies.

On behalf of the co-authors,

Will

In this document, the reviewer's comments are in black. Our responses to those comments are in red.

1) Regarding the uncertainty in the RTD of +/- 0.001 K. Is this a manufacturer's uncertainty, or is it deduced from the calibration of the RTD against each other. If the latter, then how this uncertainty was reached should be described in section 2.2.

- The uncertainty of +/- 0.001 K was determined during the calibration of the RTDs against each other. We have described the process for determining the uncertainty in Section 2.2.

2) Indeed, if it was the latter, then in line 115-116 of the track changes manuscript, this should report the manufacturer's uncertainty. And how 0.001 K was reached described in a separate sentence.

- We have reported the manufacturer's uncertainty and have described how we reached the uncertainty of +/- 0.001 K in Section 2.2.

3) I would suggest adding to the caption of Fig.6 that the uncertainty is too small to plot on the graph.

- We have added "The uncertainty in the temperature is too small to be seen on the graph." to the caption of Fig.6.

4) Since time is an important component of this study, it would help to clarify that the quality of the fit in Fig. 6 does not change with time. Can that be clarified in the manuscript.

- We have added "The goodness of the fit does not change over time." to the caption of Fig.6.

5) Can the description of the calibration of the sonic temperature sensor be added to section 2.2, or alternatively a reference, if this has been done before in a previous study for the Pi Chamber.

- We have included a description of the calibration procedure to Section 2.2 of the manuscript.